# Prioritization of New Candidate Genes for Rare Genetic Diseases by a Disease-Aware Evaluation of Heterogeneous Molecular Networks

**DOI:** 10.3390/ijms24021661

**Published:** 2023-01-14

**Authors:** Lorena de la Fuente, Marta Del Pozo-Valero, Irene Perea-Romero, Fiona Blanco-Kelly, Lidia Fernández-Caballero, Marta Cortón, Carmen Ayuso, Pablo Mínguez

**Affiliations:** 1Department of Genetics, Health Research Institute–Fundación Jiménez Díaz University Hospital, Universidad Autónoma de Madrid (IIS-FJD, UAM), 28049 Madrid, Spain; 2Center for Biomedical Network Research on Rare Diseases (CIBERER), Instituto de Salud Carlos III (ISCIII), 28040 Madrid, Spain; 3Bioinformatics Unit, Health Research Institute–Fundación Jiménez Díaz University Hospital, Universidad Autónoma de Madrid (IIS-FJD, UAM), 28049 Madrid, Spain

**Keywords:** rare diseases, genetic diagnosis, sequencing, candidate gene prediction, variant prioritization, network biology

## Abstract

Screening for pathogenic variants in the diagnosis of rare genetic diseases can now be performed on all genes thanks to the application of whole exome and genome sequencing (WES, WGS). Yet the repertoire of gene–disease associations is not complete. Several computer-based algorithms and databases integrate distinct gene–gene functional networks to accelerate the discovery of gene–disease associations. We hypothesize that the ability of every type of information to extract relevant insights is disease-dependent. We compiled 33 functional networks classified into 13 knowledge categories (KCs) and observed large variability in their ability to recover genes associated with 91 genetic diseases, as measured using efficiency and exclusivity. We developed GLOWgenes, a network-based algorithm that applies random walk with restart to evaluate KCs’ ability to recover genes from a given list associated with a phenotype and modulates the prediction of new candidates accordingly. Comparison with other integration strategies and tools shows that our disease-aware approach can boost the discovery of new gene–disease associations, especially for the less obvious ones. KC contribution also varies if obtained using recently discovered genes. Applied to 15 unsolved WES, GLOWgenes proposed three new genes to be involved in the phenotypes of patients with syndromic inherited retinal dystrophies.

## 1. Introduction

A usual first step in biomedicine is now to use omics to provide a first bunch of hypotheses on gene or protein associations with phenotypes that are, afterward, prioritized for further exploration. In this sense, rare diseases (RDs) are probably the paradigm in the application of bioinformatics tools for the prioritization of gene–disease associations. They are mainly Mendelian genetic diseases, and, in their diagnosis, it is now possible to screen the whole coding region or the entire genome using whole exome and genome sequencing (WES and WGS) in search of causative variants. This extracts, for every patient, a large number of candidate variants beyond the genes already associated with the disease [1], including not only pathogenic mutations but also variants of uncertain significance and variants with conflicting interpretations. Their diagnosis needs to be boosted in order to increase the low ratio of solved cases [2]. The challenge is to overcome the small cohort sizes and the low percentage of cases expected to be explained by unknown disease-causing genes.

Under this scenario, several computational disease–gene association prediction methods and databases have been developed to help in the prioritization of candidate genes. They can be divided into two general classes: (1) seed-based methods, which do not have *a priori* information about the interrogated trait but the functional landscape of a set of genes (seeds) provided by the user [3,4,5,6,7]; and (2) predefined disease methods [8,9,10], which initiate the hunt from terms defining the disease. While seed-based algorithms are more flexible in the phenotype definition, predefined-disease methods take advantage of the knowledge accumulated for the pathologies. For both, the strategy for extracting new associations can also vary [11] from using text-mining techniques (DisGeNET [8], DISEASES [9], ENSEMBLE [12]), network-based methods (ToppNet [4], GUILDify [5], SNOW [13]), machine learning methods (ProDiGe [14], Phenolyzer [10]), or algorithms based on functional similarity (Endeavour [3], ToppFun [4]). They have in common the ability to screen large datasets to extract hidden associations. These sources can be of different natures, ranging from literature to different flavors of omics data, including their integration [5,7], which has been reported to increase accuracy [15,16]. On top of this, resources such as PanelApp [17] provide manually curated candidate genes for genetic diseases based on community feedback.

A major challenge in the application of this type of resource to the diagnosis of RDs involves strengthening the discovery in both diseases poorly studied, so, lacking a solid background of knowledge, as well as in diseases where the missing genes have less obvious relationships to the known repertoire. With this goal in mind, we compiled several types of datasets with gene and protein annotation and tested their ability to retrieve relevant gene–disease associations on many RDs using network biology. With the conclusions, we developed a seed-based algorithm, called GLOWgenes, which is able to adapt its performance to every queried phenotype as well as to potentiate the extraction of less obvious associations. GLOWgenes has been benchmarked against current available tools, and it has been implemented to work integrated into a variant calling pipeline in the diagnosis of RDs.

## 2. Results

### 2.1. A Compilation of Heterogeneous Gene–Gene Functional Association Networks

We wanted to build a diverse and rich gene functional information framework to be used for the prediction of new gene–disease associations. Thus, we compiled 33 publicly available sources with distinct functional information about human genes (Appendix A). Of them, 22 are in a gene–gene network format [18,19,20,21,22,23,24,25,26,27,28,29,30,31], and 11 are gene annotations [32,33,34,35,36,37,38,39] that were transformed into co-annotation networks, with genes as nodes and pairwise relationships as edges (see Methods). When available, we coded the strength of the gene–gene associations using weighted edges (Appendix A). We classified the 33 networks into 13 knowledge categories (KCs) covering different aspects of cell regulation and knowledge generation (Figure 1A, Appendix A). The KCs include: (i) gene co-citation in literature, (ii) coessentiality as genetic interactions, (iii) co-expression, (iv) colocalization in cellular organelles, (v) protein complexes, (vi) targets drug sharing, (vii) shared gene functional annotation, (viii) features from genomic localization throughout evolution, (ix) shared phenotypic annotation from mouse models, (x) participation in molecular pathways, (xi) shared human gene phenotypes, (xii) protein–protein physical interactions (PPIs), and (xiii) regulation of gene expression. Regarding the number of sources in each KC, co-essentiality with five datasets is the KC with more contributing sources, and STRING [18], the database that contributes more datasets spread over five KCs, is in two of them alone (co-citations and genomic features). The networks have different sizes and shapes as shown by the number of nodes, edges, and clustering coefficient (Figure 1B). The regulatory database RegNetwork [30] has the highest number of nodes, while data provided by CRISPR screenings have the largest number of associations. The co-citation network from STRING stands out with the second largest number of nodes and edges, having a low clustering coefficient, which suggests a hub-based connectivity. A grouping based on the overlap coefficients recovers the functional associations of the networks fitting quite well with our classification into KCs (Appendix A).

### 2.2. KCs Show Different Capabilities for the Recovery of Gene-Phenotype Associations

In a guilt-by-association approach, functional networks are a useful tool to associate genes with new functions or phenotypes based on their neighborhood [40]. We aimed to assess the capability of the different KCs to recover the information from gene–disease associations. To perform this assessment, we selected 91 gene sets (or panels) used in the diagnosis of genetic diseases, classified into 20 disease families, from the PanelApp resource [17] (see Methods, Appendix A) and applied a random walk with restart (RWWR) model [41] to every network using a training subset with 70% of the genes. The remaining genes of the gene sets (30%) were used to validate the prediction capability of each network by calculating two parameters: (1) efficiency (recall), reflecting the true associations caught between genes and phenotype and (2) exclusivity, or capacity to recover genes others cannot, measured as the mean gene specificity [42]. Both parameters are represented as the average of a 20-fold performance. For every KC, we select a single individual network as the best representative based on its area under the precision–recall-gain analysis (AUPRG) [43] (see Methods). This selection varies for every disease, particularly in the regulation networks (Appendix A).

To compare KCs, we calculated the efficiency and exclusivity of their best-performing networks at the n-top, taking n as the number of genes in the input disease panel. In a general view, KC efficiency varies substantially even within diseases of the same family (Figure 2A). In Figure 2A, KCs are sorted by their overall performance. Generally, phenotype and co-citation KCs achieve the best efficiency, although all KCs show, alternatively, a high relevance in particular diseases (Appendix A). Some KCs work well in specific disease families: complexes in ciliopathies (Wilcoxon signed-rank test, WST, *p*-value < 0.001), drug sharing in tumor syndromes (WST, *p*-value < 0.001), or co-essentiality in metabolic disorders (WST, *p*-value  < 0.001) (Figure 2A and Appendix A). Remarkably, there is a high intrafamily disease variation with KCs outperforming the general trend for particular diseases (Appendix A). A focus on the efficiency for four distinct diseases (pilot diseases) accentuates the differential importance of specific KCs (Figure 2B).

We also calculated the exclusivity of KCs recovering gene–disease associations at n-top to measure their capacity to detect genes uniquely. The scenario here is much more diverse than that observed with the efficiency; there are no KCs that concentrate the exclusivity, but all are important in almost every disease family (Figure 2C and Appendix A). Interestingly, some KCs with low and medium overall recall, such as regulation or mouse phenotypes (Figure 2A), increase their importance in exclusivity having third and fourth best general performance, respectively (Figure 2C). On the other hand, in terms of exclusivity, other KCs are globally downgraded compared to efficiency performance, as is the case of PPIs, falling from fourth to 11th place (Figure 2C).

Figure 2D displays a close look at the efficiency and exclusivity of KCs in the recovery of genes associated with our four pilot diseases. We observe no clear correlation between efficiency and exclusivity, but instead we found several KCs with very low efficiency but high exclusivity.

### 2.3. The History of Acquired Knowledge on a Disease Influences KCs Contribution to the Recovery of Disease-Associated Genes

Since the discovery rate of gene–disease associations can vary over the years depending on several factors (disease prevalence, genetic heterogeneity, or scientific/monetary efforts), we hypothesize that the accumulated knowledge on each disease at the time of analysis may be also an aspect to be taken into account in the gene discovery process. Thus, we tested whether the performance of KCs in recovering genes for a particular disease changes if time is also considered. Thus, using 246 gene sets describing diseases and phenotypes from DisGeNET [44] (see Methods), we compared KC performance in recovering genes using: (1) 30% of the genes chosen randomly in a 20-fold cross-validation and (2) genes recently discovered (time-aware validation) (Figure 3A). Comparisons were made using an integration score that considers efficiency and exclusivity (see Methods).

Using the time-aware approach, we observe a strong decrease in the performance of the KC co-citation (WST, *p*-value = 3.21 × 10^−25^) (Figure 3B). The same trend is reported for the KCs’ functional annotation and drug sharing (WST, *p*-values = 5.22 × 10^−5^ and 2.71 × 10^−9^, respectively). On the other hand, other KCs such as regulation and co-essentiality increase their capacity to detect recent gene–disease associations (WST, *p*-values = 4.42 × 10^−23^ and 1.53 × 10^−14^, respectively). To illustrate these results, we selected the four DisGeNET gene sets with the most dramatic changes in KC performance (Figure 3C and Methods). As shown above, co-citation displays a severe decrease in its performance for all four diseases using the time-aware validation. For the rest of the KCs, the patterns are highly variable. Thus, genes recovered for attention deficit hyperactivity disorder seem to be caught mainly by KCs co-expression, co-essentiality, colocalization, and drug sharing. In coronary heart disease, with a validation cut-off at year 2012, co-citation continues to play a predominant role followed by protein complexes and colocalization. In prostate carcinoma, coessentiality, regulation, and canonical pathways information overtake co-citation, whose influence using validation on random genes was 50% of the total. The case of toxic hepatitis is especially striking, with the new pattern conserving the same trend but with a significant increase in regulation as KC with more potential to detect late insights (Appendix A).

### 2.4. A Disease-Aware Algorithm to Integrate Several Sources of Knowledge

As the contribution of KCs in the recovery of causing genes is disease-specific, we developed a novel algorithm, named GLOWgenes, which adapts its performance to the special capabilities of every KC to recover genes associated with specific phenotypes. Thus, given a gene set associated to a particular disease (GAD), or any trait of interest, GLOWgenes applies two steps in parallel (Figure 4): (1) evaluates performance of each network (N = 33) in recovering genes from the gene set (step repeated 20 times), where the algorithm performs a RWWR for all networks using 70% of the genes as seeds, calculates KC weights based on their efficiency and exclusivity using the rest of the genes from the gene set (test genes), and chooses the network with best performance of each KC. If provided, seeds and test genes can also be defined using a cut-off defined by the year of publication of the gene-disease associations; and (2) for every KC (using the best network from each selected in step 1), apply RWWR on the whole set of GADs to rank all genes. Finally, GLOWgenes uses KC weights (step 1) to modulate the gene rankings (step 2) and produce a single gene ranking, in which genes in the GAD have a value of zero, and the rest are sorted in descending order with value 1 as the most associated.

We compared our disease-aware integration strategy with two widely used alternative approaches for disease candidate gene prediction (Figure 5A): (1) integration of datasets into a composite network prior to gene prioritization [15,45,46] and (2) combination of multiple rank lists using order statistics [3,47,48], represented by two methods, data fusion [47] and robust rank aggregation [3,48]. We applied these three methods, and ours, to disease gene sets from PanelApp. To evaluate the predictive capability in a gene-discovery scenario, we used their collection of candidate genes, called red genes (RGs), as validation sets. A total of 63 PanelApp gene sets, with more than 10 RGs, were used. Recall was calculated at various n-top genes. Using the same input information for all four algorithms (see Methods), the results show that our disease-aware integration approach achieves the best average performance across all disease gene sets, with a mean AUPRG = 0.92 (Figure 5B) outperforming the rest of the data integration approaches together in 52% of the diseases, looking only at the first 16 predicted genes (n-top = 16), and in 80% of the diseases considering the first 256 genes (n-top = 256) of the rankings (Figure 5C). The composite network performed second in the global ranking with 10 and 32 diseases with the best and second-best recalls, respectively, considering the same number of genes as the validated set (Appendix A).

### 2.5. Comparative Assessment of Disease Candidate Gene Prioritization Methods

To extend the evaluation of GLOWgenes beyond the integration method, we compared its performance against other tools that predict disease–gene associations using a list of genes as seeds and do not need a preselection of candidates, called here seed-association (SA) methods. In addition, we introduced as external references, methods that predict associations using disease terms instead of gene lists, called here predefined disease (PD) methods [9,44,49]. In total, we selected six SA methods, including: DIAMOnD [6], Guildifyv2.0 [5], ToppGenet-net [4,50], ToppGenet-fun [4,51], Endeavour [3], and GeneMANIA [7], and three PD methods, including: DISEASES database [9], BEFREE [44,49] (as extracted from DisGeNET), and predictions from DisGeNET (see Methods, Appendix A). First, we tested the ability to recover genes with well-supported evidence in their association to the disease, represented by the PanelApp Green–Amber validation genes (GAG) in our four pilot diseases. Figure 6A shows that PD methods overall outperform SA methods in all four diseases with better recall levels at a different number of top genes selected. Only GLOWgenes has similar or better recall (e.g., in cardiomyopathies) than DisGeNET, BEFREE, and DISEASE predictions, and performs widely better than its counterpart methods, with only ToppGene [4] reaching similar recall levels in severe microcephaly (Figure 6A).

Next, we carried out a similar benchmark using PanelApp RGs of the same diseases as a proxy to test the ability to catch new gene–disease associations beyond the current knowledge. We observe a general decrease in the recall obtained by all methods, with SA and PD methods performing similarly (Figure 6B). In this scenario, GLOWgenes outperforms all other methods in three out of four evaluated diseases and achieves similar recall levels for hearing loss. After GLOWgenes, BEFREE, DisGeNET, GeneMANIA, ToppGene, and Endeavour ranked in alternate positions. GLOWgenes is also the method that captures more unique genes (Appendix A). Among all SA tools considered, only three of them provide programmatic access options for automatic task execution (NetCombo from GUILDIFY2.0, DIAMOnD, and GeneMANIA). We compared their performance using RG validation sets for 63 PanelApp disease gene sets, with GLOWgenes obtaining the best recall across all diseases globally (Appendix A).

### 2.6. Gene Discovery in Undiagnosed Cases with Syndromic Inherited Retinal Dystrophy Using GLOWgenes

A major application of gene-phenotype discovery methods is to propose candidate genes to be implicated in RDs. GLOWgenes is adapted to annotate a list of variants provided in a table format as it is coupled to our reanalysis pipeline [52]. Here, we have applied GLOWgenes for the discovery of candidate genes involved in syndromic forms of inherited retinal dystrophies (sIRD). As a proof of concept, 15 unsolved cases with sIRD and WES data available from the cohort of the Fundación Jiménez Díaz University Hospital (FJD-UH, Madrid, Spain) [2] were analyzed. These samples were previously analyzed at the FJD-UH with no variants found fitting the phenotype in known sIRD-associated genes (sIRD-virtual-panel with 198 genes used in the diagnostic protocol for sIRD, Appendix A). After variant calling [52] and filtering by: (i) quality (Q = 100, DP = 10), (ii) predicted pathogenicity (CADD > 15), and (iii) allele frequency (gnomADg_AF_POPMAX < 0.05/NA), we obtained an average of 1562 variants per sample in around 1000 genes without previous association to sIRDs. GLOWgenes was run using the sIRD-virtual-panel as seeds to provide a gene ranking that was used to prioritize the filtered variants of each case.

Thus, by interrogating the top 300 predicted gene list (Table 1), we yielded a total of three candidate genes with pathogenic variants that fitted the phenotype in two patients with sIRD. We found two heterozygous variants in Case 1 (Table 1). First, we found a pathogenic missense variant in the *SHH* gene previously reported to cause holoprosencephaly type 3 (MIM:142945), microphthalmia with coloboma type 5 (MIM:611638), schizencephaly (MIM:269160), or single median maxillary central incisor (MIM:147250) and with no clear mode of inheritance. Second, we found a heterozygous pathogenic stop-gain variant in the *DNAH5* gene, a dynein involved in primary ciliary dyskinesia type 3, with or without situs inversus (MIM:608644) and reported as autosomal recessive.

In a second case (Case 2, Table 1), we found a heterozygous likely pathogenic variant in the *GLI1* gene that has been associated with two types of polydactyly (phenotype present in our patient) and reported as autosomal recessive (source: OMIM database).

Regarding the contribution of KCs in the prioritization of the three genes proposed, reporting only those reaching >50% of the total score (Appendix A), *SHH* was found mainly by the signal provided by co-citation (46%) and mouse models (22%) networks, *DNAH1* by the co-citation (33%), co-expression (29%), and mouse models (19%) networks, and *GLI1* by the co-citation (26%), PPIs (19%), and regulation (10%) networks.

## 3. Discussion

Around 50% of patients with RDs do not have an accurate diagnosis [53], and there is an urgent need to reduce this gap [54]. One limitation is the lack of certainty about the clinical significance of the genomic variation data [1], with 40% of the variants annotated as having uncertain significance in the ClinVar database [54]. If all genes were suspicious, a regular WES analysis may detect around 20,000 single nucleotide variants [55], 200 of them very rare (MAF < 0.1%), and an average of 30 not detected in any other individual [56]. All these numbers describe a scenario in which the use of WES and WGS to diagnose RD patients with causative variants outside the current knowledge needs the application of prioritization algorithms. Current major national and international initiatives focus on the use of these two sequencing techniques (WES and WGS) to assume discovery [56].

Methods for the prediction of new gene–disease associations evolved from those developed for predicting gene and protein function [57,58] with social network [59] and data integration at the foundations [60]. The needs are now focused on the development of specific and accurate resources that overcome the limitations of every area of application. Here, RDs represent a rather particular challenge. There are over 7000 diseases with different levels of genetic and phenotype heterogeneity. They are mainly Mendelian, so if the variant is in the coding DNA, we search for a single gene. Moreover, they generally have small cohorts available, so a clear genotype–phenotype correlation can be difficult to find. Thus, RD diagnosis can become a “needle in a haystack” task where the solution is still far connected to current knowledge. In consequence, to increase their efficiency in difficult cases, gene–disease prediction methods are encouraged to use new and various types of functional annotations and combine them accurately [15,16,61,62]. In this study, we first confirmed that different types of gene–gene functional association networks had different capabilities in recovering known genes associated with a large collection of RDs. In terms of efficiency, the co-annotation network using phenotypes, as well as the co-citation network using STRING text-mining data, are the best performing sources in general, with particular diseases having alternative important sources. The high accuracy of the phenotype-based networks was expected as HPO terms are close in definition to rare diseases. However, differences are especially evident when considering genes detected uniquely by one source, which we called exclusivity (Figure 2). This fact might be of essential importance considering its application to Mendelian diseases.

With needs, limitations, and preliminary results in hand, we developed GLOWgenes (www.glowgenes.org), a seeds-based algorithm to prioritize candidate genes with the main application in the diagnosis of RDs using WES and WGS. GLOWgenes applies the guilt-by-association principle using the RWWR method on multiple networks as different layers of functional information [62,63,64], preserving not only the nature of the associations but also the topology of the networks. GLOWgenes is unique in the sense of evaluating the performance of different sources in every case to assign weights that are afterward integrated. We tested our integration method using well-known representatives of this kind of approach with good results (Figure 3).

A second benchmark was performed comparing GLOWgenes with state-of-the-art tools to predict gene–disease associations based on seeds. Here, popular PD tools were used as external references. Using well-known gene–disease associations as testers (GAGs), PD methods achieved more prediction power than SA algorithms, with GLOWgenes behaving better than SA tools and similarly to PD tools (Figure 6A). When considering a more realistic discovery scenario, that is, testing the ability to recover candidate genes (RGs), the difference in the performance of SA and PD methods is less evident, with GLOWgenes also at the top of performers (Figure 6B). The main difference between PD and SA tools is that PD methods cannot be easily adapted to new, or a combination of, phenotypes as required in genetic syndromes, for instance [65]. In fact, most PD algorithms are based on text-mining approaches that reward with a good recall but limit their ability to go beyond current knowledge. In this work, we provide two results that suggest that text-mining may mask interesting RD candidate genes. First, we observed a dramatic reduction in the importance of text-mining when trying to highlight recently discovered genes, using our time-aware partitioning strategy compared to recovering randomly selected genes (Figure 3). Second, if efficiency is measured in RGs, the methods based on text-mining, mainly PD methods, reduce their performance (Figure 6B).

Another interesting finding is that, in most cases, SA methods aggregating several functional sources (GLOWgenes, GeneMANIA, ToppGene, and Endeveour) performed better than network-based approaches using only PPI information (Guildifyv2.0, DIAMonD, ToppNet) (Appendix A). In general, we observe that GLOWgenes behaves well under any scenario tested, being also the method catching more genes uniquely. Caveats specific to these types of approaches include the low overall recall of the algorithms and the accuracy, variety, and level of coverage of the available functional gene and protein associations. In their application to the diagnosis of RDs, it must be taken into account that the causal variants may not be in the coding DNA.

The initial implementation of GLOWgenes that we present here is designed to be coupled to a pipeline of variant calling and annotation of DNASeq data. GLOWgenes is now serving candidate genes in the research of RDs in the FJD-UH annotating variants using our reanalysis FJD-pipeline [52]. Although its first motivation is to contribute to the genetic diagnosis of RDs, GLOWgenes is a predictor of new gene–disease associations that can help in other discovery scenarios. For instance, it has also been used in the search for genomic variants that provide susceptibility to COVID-19 [66]. As a proof of concept, we provide here the results of the analysis of 15 WES of patients with sIRD. Three genes are reported here to provide new insights in two sIRD cases. In the first case, we found two pathogenic variants in heterozygosis involving two genes: *SHH* and *DNAH5*. Regarding the *SHH* gene, mutations produce a related phenotype, and it has a role in the neurogenesis of the zebrafish retina [52]. Although the proband’s father carries the same variant, *SHH* mutations are described to have a high variability of expression [67]. On the other hand, *DNAH5* has been proposed as a candidate gene for retinal dystrophies using a different method based on the aggregation of variants in unsolved cases compared to controls [1]. In a second case, we found a monoallelic variant in the recessive gene *GLI1*, which explains partially the patient’s phenotype [68]. Further analyses are needed to reach a conclusive diagnostic in both cases.

In conclusion, this work describes the special needs and limitations in the prioritization of candidate genes in the diagnosis of RDs using WES and WGS and presents a method that is hopefully useful to reduce the diagnostic gap of this type of patient.

## 4. Materials and Methods

### 4.1. Compilation of Gene-Gene Functional Associations from Public Sources

We gathered functional gene or protein information from various sources [18,19,20,21,22,23,24,25,26,27,28,29,30,31,32,33,34,35,36,37,38,39] to build 33 networks of functional gene–gene interactions (Appendix A). The sources with protein information had no isoform resolution, so a one-to-one correspondence with the genes can be assumed. Networks were classified into 13 classes, named here as knowledge categories (KCs) according to the nature of the functional information that they contain (Appendix A). Gene identifiers from sources were all mapped into HGNC gene symbols, excluding interactions involving genes without HGNC mapping. Weighted edges were kept and normalized. If unavailable, edge weights were all set up to 1 (Appendix A). The network named HPOext is the resultant network of imputing genes to HPO ancestors from their more specific annotations as provided by HPO. Gene annotations not represented as gene or protein relationships were transformed into gene–gene co-annotation networks, describing genes (nodes) and their functional relationship (edges). Thus, for networks describing human gene–phenotype similarity using Human Phenotype Ontology (HPO) terms [32] and phenotype similarity using mouse orthologs from the Mouse Genome Informatics (MGI) [16], we calculated Jaccard similarity for each pair of genes sharing at least one HPO term and constructed a null distribution of Jaccard values to compute z-scores. Significant gene interactions (z-score > 1.96; *p*-values < 0.05) were selected to generate the network of phenotypes. Z-score values were used as edge weights. Direct linkage between genes sharing annotations was used for network construction for the drug–gene interaction database (DGIdb) [34] and complexes from CORUM [36]. The co-expression network from COXPRESdb [19] was created considering gene pairs with mutual rank (MR) co-expression correlations under 2000. The inverse of the MR was used as edge weights. The ProteomeHD [24] co-expression network was filtered to contain only the top-scoring 0.5% pairwise gene co-regulations, as recommended. Co-essentiality networks were derived from: (i) inferred genetic interactions from CRISPR screens by Rauscher et al. [27], (ii) weighted co-essentiality gene interactions obtained by Kim et al. [29] using CRISPR essentiality screens from the Avana Project [31]; (iii) gene fitness rank correlation coefficients calculated using two different RNAi- and CRISPR-Cas9-based screening datasets and following Pan et al. [35] processing pipeline (rank threshold = 1024; inverse rank threshold as edge weights); and (iv) genetic interactions from the BIOGRID database [25]. Networks derived from GO annotation were constructed using semantic similarity from GOGO [37] for gene pairs sharing at least one GO term. Semantic similarity scores were used as edge weights. DoRoThea evidence levels were transformed into numeric values for regulatory network weighting normalized to 0–1 range.

### 4.2. Comparison of Network Topology and Similarity Measurement

Attributes from each compiled network (node size, number of edges, average clustering coefficient) were calculated using the python package NetworkX.

To compare the different networks, we computed edge-wise network similarity using the overlap coefficient (OC), also known as the Szymkiewicz–Simpson coefficient. OC is defined as the edge size of the union of two graphs (G1 and G2) over the size of the smaller set between G1 and G2.

### 4.3. Disease-Associated Gene Sets Used in the Evaluation of KCs and Benchmarks

Assessment of KCs’ importance in gene recovery was performed using disease-associated gene sets with high and moderate evidence (Green and Amber genes, GAGs) from the Genomics England PanelApp [17]. Disease-associated genomic regions not fitted to genes such as copy number variations (CNVs) and short tandem repeats were filtered out from the panels. We grouped the panels into disease families using the level 2 provided in the disease classification by PanelApp. Panels lacking classification were manually curated by clinicians at FJD-UH. Although we observed that disease families might partially overlap in their definition, for instance with ciliopathies and ophthalmological disorders, we kept the PanelApp definition.

Alternative integration strategies were benchmarked using PanelApp red genes (RGs), defined as disease-promising genes that need further evidence for clinical disease diagnosis. Gene–disease prediction tools were benchmarked using both GAGs and RGs.

When PanelApp gene sets were used, super panels (involving a mix of phenotypes) were excluded. In the case of using GAGs, only panels with at least 40 GAGs were selected (91 panels). When RGs were used, only panels with at least 10 RGs were considered (70 panels).

### 4.4. Network Signal Propagation

Network propagation of signal (trait or disease annotated) provided by gene sets was performed using the random walk with restart (RWWR) model proposed by Kohler et al. [41] as implemented in python software NetworkX and modified to generate a degree weighted adjacency matrix for subsequent propagation in order to minimize node degree bias (avoid bias towards highly connected genes). The modification was performed following the edge weight transformation proposed by Vanunu et al. [69] and used by others [70,71]. Restarting walk probability (rwp) was set up to 0.75, as suggested by Vanunu et al. [69]. Convergence is decided when the probability difference between two consecutive time steps is less than 10^−6^. Upon convergence, the frequency with which each node (gene) in the network has been visited is returned as a propagation score vector (S), which represents the probability of genes being associated with the input gene set. All networks were treated as weighted, if information was available, so adjacency matrices were defined providing edge weight attributes.

### 4.5. Systematic Evaluation of Evidence Networks Representing Heterogeneous Knowledge Categories

To assess the performance of a network to prioritize phenotype-associated genes, we performed a 20-fold random sub-sampling validation. Each phenotypic gene set was randomly split into 70%/30% training/test subsets. Training genes (seeds) were then propagated in each individual network using the RWWR model, and the resulting gene ranking was evaluated on test genes. The validation results were averaged over the 20 cross-validation rounds. Several metrics were calculated to assess the performance of a given network in a particular disease. Given the high imbalance nature of our dataset (high negative/positive instance ratio), the overall ranking was evaluated using the precision–recall analysis. We calculated the Precision–recall-gain [43] curve and its associated area (AUPRG). The network with the highest AUPRG in each KC is selected as the representative KC network and used for further analysis. To emphasize the top-ranked genes, instead of testing true/false separation in the total gene ranking, we also computed recall and gene specificity at different top-k thresholds. Recall evaluates the efficiency of networks to prioritize disease genes, while gene specificity is informative of their degree of detection by particular KCs. Gene specificity was calculated for every true positive gene captured as in Martinez et al. 2008 [42]. High specificity genes represent genes captured by a single KC while low specificity genes refer to genes detected by all KCs. The overall KC specificity level was calculated by averaging gene specificity scores and summarizes the ability of a KC to detect disease-specific genes, named here as exclusivity.

### 4.6. Time-Aware Network Evaluation

The time-aware network evaluation analyses whether the contribution of KCs in gene recovery for a particular phenotype changes if time is also considered. As input, the publication year for each gene associated with the phenotype under study must be provided. GLOWgenes takes that time-sorted list of genes and evaluates each individual source of evidence (network) using a time-aware validation, which relies on gene publication year to split gene sets into 70/30 train/test partitions. The oldest genes are used as the training set to determine the ability of each network to recover the most recently discovered genes (test set). Partition publication time, metrics, and overall gene prioritization rank using time-aware mode are provided. Moreover, random vs time-aware network evaluation statistics are also computed in time-aware mode.

Gene–disease associations from DisGeNET [44] were used in the evaluation of the time-aware approach. The publication year was recovered from DisGeNET SQLite database, and time-aware disease gene sets were generated based on the UMLS Concept ID (CUI) disease classification, comprising phenotype and disease categories. We only considered gene–disease associations from curated sources (CGI, CLINGEN, GENOMICS_ENGLAND, CTD_human, PSYGENET, ORPHANET, UNIPROT, GWASCAT, GWASDB, CLINVAR) and limited the analysis to gene sets with at least 70 genes, with a total of 246 disease gene sets remaining for time-aware disease candidate gene discovery.

### 4.7. GLOWgenes Algorithm

GLOWgenes is based on the integration of gene prioritization results from *k* heterogeneous knowledge categories (KCs), called here disease-aware prioritization. Gene prioritization results from each KC representative network *i* are integrated into a unified ranking by incorporating *ad-hoc* information about their individual performance on the particular phenotype/disease under study.

The strategy involves three main steps: Step 1. RWWR propagation of *n* phenotype/disease-related genes on each individual network result in a gene ranking representing their association strength to the input gene set. Step 2. Network evaluation and selection follow the steps listed in the previous section:For each network out of the 33 considered in this analysis, we performed a 20× random cross-validation of phenotype-associated input genes (70/30 training/test). For each partition:The training subset is propagated using a RWWR model.The area under the precision/recall gain curve is calculated.Networks with the highest mean AUPRG at each KC are selected as KC representative networks and used for further analysis.KC exclusivity and efficiency (recall) are calculated at different top-k for each representative KC network.Integration scores are calculated as the product of efficiency and exclusivity for each KC as a measure of the ability to recover input disease-associated genes.

Step 3. RWWR Scores (step 1) of representative KC networks (step 2) are integrated using the integration scores (step 2). In order to allow for unbiased integration of multiple networks, gene scores from network *i* are normalized by subtracting the mean of the scores of all nodes from the score of node n and then dividing by the standard deviation of the distribution. The normalized z-scores for representative KC networks are then merged in a k-n matrix (k = KCs; n = genes) and score imputation is applied for missing values. Regarding imputation, a gene missing in the evaluation of a KC takes the minimum normalized score observed for a gene in that KC. Disease-aware integration of the k KC representative networks is performed by weighing the gene z-scores in each network using the performance factor obtained from the evaluation of *i* network and subsequent combination of ranks by averaging these weighted scores. A final overall gene ranking is generated. GLOWgenes is available at www.glowgenes.org.

### 4.8. Evaluation of Alternative Approaches for Data Integration in Gene Discovery

We compared the integration approach used by GLOWgenes (disease-aware integration) to two common strategies for network-based data integration in gene discovery: the fusion of rankings generated by individual modeling of datasets by order statistics [3,47,48] and the construction of a composite network for subsequent modeling [15,45,46]. Regarding the first, we evaluated two rank aggregation algorithms based on order statistics: (a) the optimization of the Stuart algorithm [72] by Aerts [47,63] and implemented in Endeavour [3] and (b) the Robust Rank Aggregation (RRA) algorithm [48]. We ran the implementations of both algorithms using the R package RobustRankAggreg. For the network integration strategy, we constructed a composite network using our 33 defined networks by adapting the most optimal configuration of a composite network described by Huang et al. [15]. Selection of edges supported by at least two sources from different KCs generated a composite network containing 4,509,215 edges and 19,302 nodes. Alternative strategies were run across 63 PanelApp GAG gene sets and validated in RG genes. Recall-at-k was measured to quantify what fraction of all the disease genes are ranked within the first k predicted genes (k = 8, 16, 32, 64, 128, 256, 512). The best integration methods are selected at recall-at-n (validation set size). The mean area under the precision–recall curve (AUPRG) across gene sets was also calculated for each approach.

### 4.9. Benchmark of Tools for the Prioritization of New Disease-Related Genes

For benchmarking, we selected tools that met four criteria: (a) accessible, (b) updated since 2016, (c) predicting gene–disease associations from a user-input set of genes previously linked to a phenotype (seeds), (d) no need for an implicit set of user-input candidate genes to operate. A total of six seed-association tools were selected and run on four pilot disease gene panels (described before) using their default parameters: Endeavour, which uses a whole-genome approach [3], ToppNet algorithm, which uses ToppGenet for selection of the first neighboring genes in PPI as candidates (ToppGenet—network based, ToppGenet-net) [4,50], ToppGene algorithm, which uses also ToppGenet for candidate selection (ToppGenet—functional similarity, ToppGenet-fun) [4,50], NetComb-GUILDIfy [5], the DIAMOnD [6] version implemented in GUILDIFY2.0 web server, and GeneMANIA using its own API [73]. In all cases, a ranked list of candidate genes was derived to evaluate their performance on GAGs and RGs validation sets.

We included as external references in the benchmark, pre-defined disease methods integrating gene–disease associations: DISEASES [9] and DisGeNET [44]. In the case of DisGeNET, we separate predictions/inferences from curated sources in order to avoid knowledge bias during benchmarking. We performed a recalculation of gene–disease association scores from pre-computed DisGeNET data by removing the proportional score associated to curated evidence, generating our DisGeNET-noncurated dataset. Moreover, we evaluated just DisGeNET predictions by considering only associations derived from biomedical literature using BEFREE (DisGeNET-BEFREE). DISEASES, DisGeNET-noncurated, and DisGeNET-BEFREE pre-computed datasets were interrogated for each evaluated PanelApp disease by selecting the gene–disease associations involving disease terms enriched in the corresponding PanelApp gene set (FDR Fisher-test < 2 × 10^−16^). To make the seed-association and pre-defined disease comparable, we also filtered out the computed rank list by removing genes contained in the PanelApp gene set under study (used as seeds for seed-associated methods).

The main characteristics and sources of evidence for both seed-association and pre-defined disease methods are listed in Appendix A. All methods were evaluated and compared across four selected diseases using both GAG and RG genes as validation sets. The selected diseases were cardiomyopathies including childhood onset, hearing loss, retinal disorders, and severe microcephaly. It was not feasible to use the full set of diseases, since most tools do not provide programmatic access.

Three SA methods (GeneMANIA, DIAMOnD, and NetCombo-GUILDIfy) provide programmatic access, so they were run on 63 PanelApp disease gene sets using RGs as validation sets. Mean recall and error at different top-k were calculated for each approach. The mean area under the precision–recall curve (AUPRG) was computed for GLOWgenes and GeneMANIA, but not for DIAMOnD and NetCombo-GUILDIfy, since they just returned the 500-top rank list of candidate genes.

### 4.10. Detection of Variants Fitting Patient Phenotype in GLOWgenes Candidates Using WES

We considered a total of 15 WES tests from patients with syndromic inherited retinal dystrophy (sIRD) phenotypes for clinical application of GLOWgenes. All cases have an inconclusive genetic diagnosis after clinical analysis of pathogenic variants at known sIRD-associated genes. We used an in-house pipeline for the detection and annotation of germline variants [52], available at https://github.com/TBLabFJD/VariantCallingFJD. Variants were filtered by quality (Q = 100, DP = 10), predicted pathogenicity (CADD > 15), and low allele frequency (gnomADg_AF_POPMAX < 0.05 or NA). Selected pathogenic WES variants were prioritized using GLOWgenes sIRD ranking of candidate genes. To generate this sIRD ranking of candidate genes, we ran GLOWgenes using as seeds the sIRD virtual gene panel used for diagnosis at FJD-UH, which contained 198 genes. Clinical diagnosis was re-assessed by analyzing GLOWgenes detected sIRD candidate genes that contained pathogenic variants in WES cases and considering information on the familiar phenotypic pattern and variant segregation, when available. Genes detected by GLOWgenes that overlapped sIRD phenotype were classified as phenotype-matched candidates and selected for further studies.

### 4.11. Relative Contribution of Each KC to the Association to Known sIRD Genes

The relative contribution of each KC to the association with known sIRD genes was calculated for each WES-supported novel sIRD candidate and represented in pie charts using the GLOWgenes-normalized RWWR integration score for each KC representative network before KC integration by GLOWgenes.

## Figures and Tables

**Figure 1 ijms-24-01661-f001:**
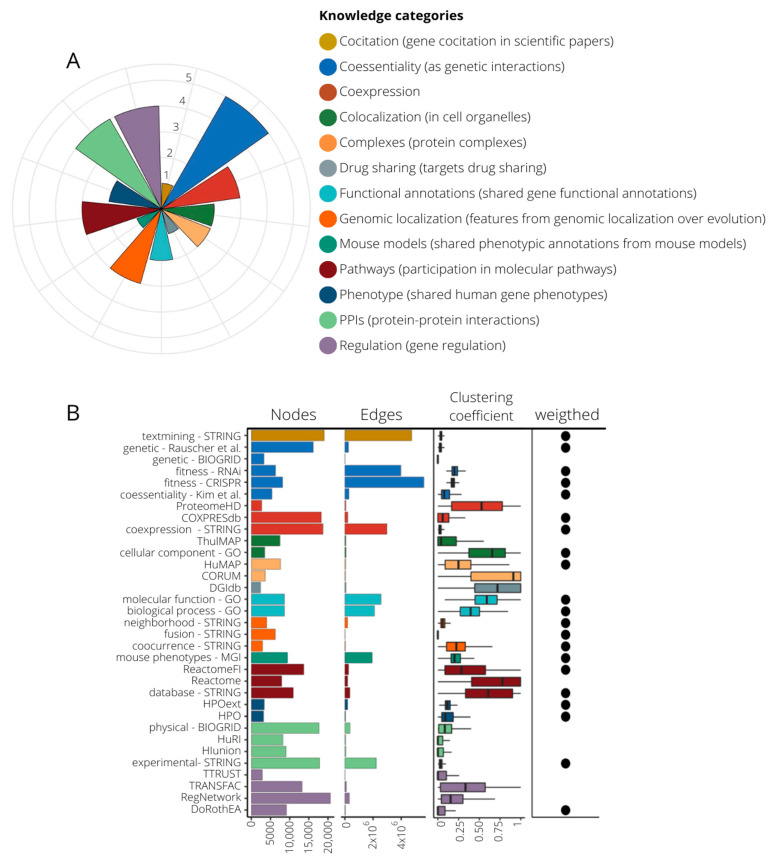
Composition of knowledge categories (KCs) and their network properties. (**A**) 13 KCs grouping 33 datasets. The pie chart indicates the number of source contributors per KC, ranging from one (drug sharing, co-citation, and mouse models) to five (co-essentiality). (**B**) Network attributes of the 33 networks generated from the compiled datasets. Here we present number of nodes, number of edges, the mean of their clustering coefficient, and whether edges are weighted or not. Color code is assigned according to their KC.

**Figure 2 ijms-24-01661-f002:**
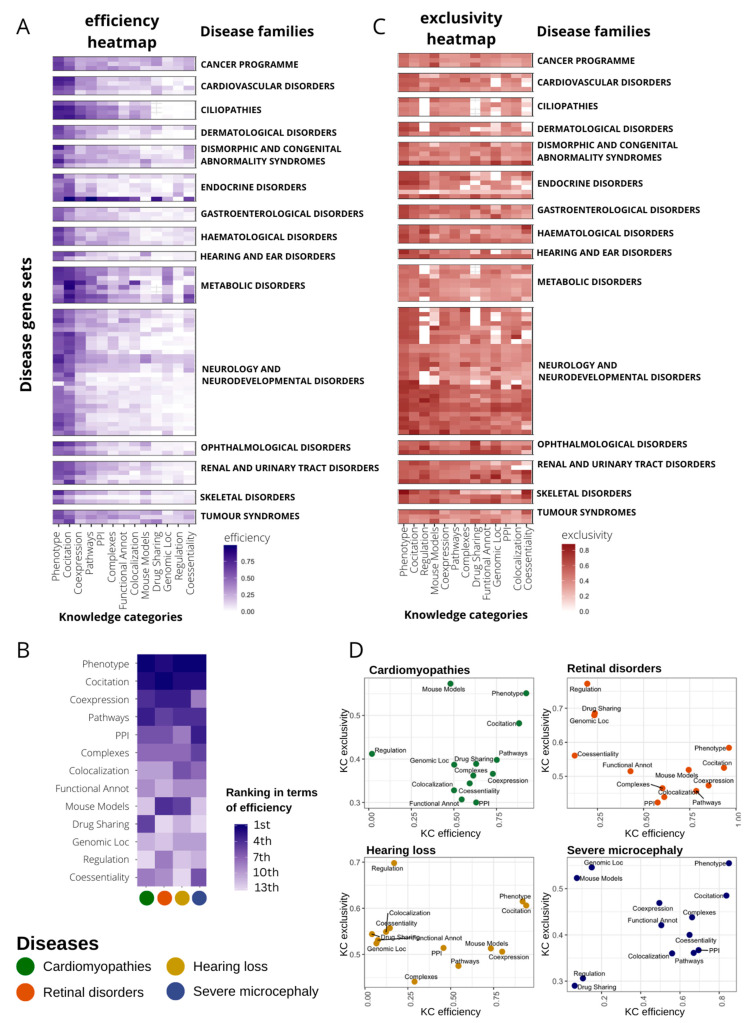
Disease-dependent performance of heterogeneous knowledge categories (KC) to recover gene–disease associations. (**A**) Heatmap representing the efficiency of every KC (x axis) in recovering genes in 91 disease gene sets from PanelApp (y axis). Efficiency is measured as the recall at top n (gene set size). Only genes with high and moderate evidence status at PanelApp (Green and Amber) were considered. Diseases are classified into families. Disease families with only one gene set were discarded for plotting purposes (seven gene sets). Diseases within families are sorted according to their similarity in the efficiency pattern using hierarchical clustering. KCs were also sorted according to median recall levels across disease gene sets. (**B**) Ranking of KC efficiency for four selected diseases (pilot diseases). (**C**) Heatmap representing the exclusivity of every KC (x axis) in gene recovery in 91 disease gene sets from PanelApp (y axis). Exclusivity is calculated as the normalized value of the mean gene specificity of genes at top n (gene set size). For representation purposes, disease filtering and plot layout were performed as in (**A**). (**D**) Scatter plot representing KC efficiency versus KC exclusivity for four selected diseases.

**Figure 3 ijms-24-01661-f003:**
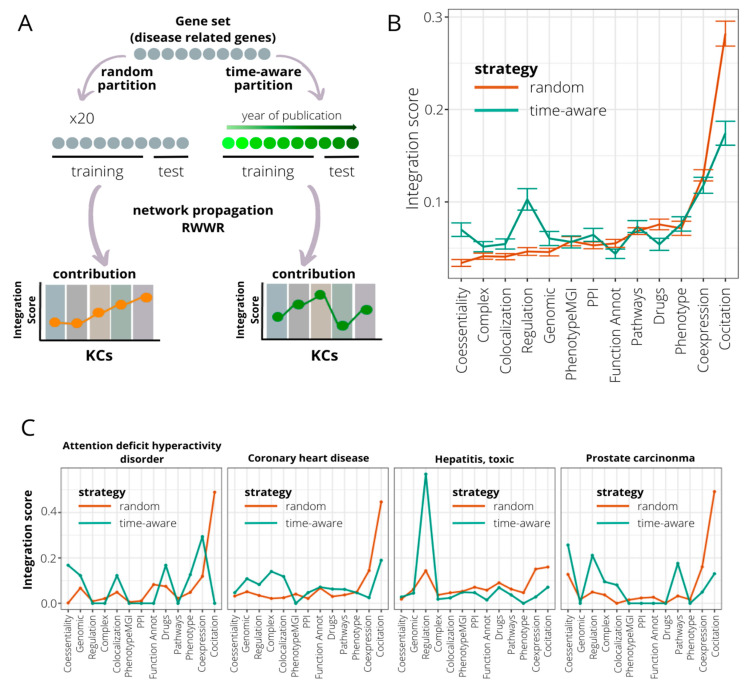
Time-aware evaluation of the contribution of knowledge categories (KC) in recovering genes associated with genetic diseases. (**A**) A time-aware evaluation approach is compared to an evaluation based on a random partition of the initial gene set in assessing the impact of KC contribution in the gene recovery associated with genetic diseases. Time-aware evaluation consists of dividing the gene set into training and testing subsets based on the year in which genes were associated with disease. The training set is composed by the older genes, and the testing subset by the newer. The KC importance is measured using the integration score, calculated as the product of efficiency and exclusivity of KC. (**B**) Comparison of integration scores when considering random (brown line) and time-aware (green line) approaches for 246 curated disease/phenotype gene sets extracted from DisGeNET. Mean recall levels and error bars are represented. KCs are sorted based on median integration scores at random evaluation. (**C**) Four disease gene sets ranked in the top 10 when sorted by additive KC variation. Integration scores for random (brown line) and time-aware (green line) approaches are shown.

**Figure 4 ijms-24-01661-f004:**
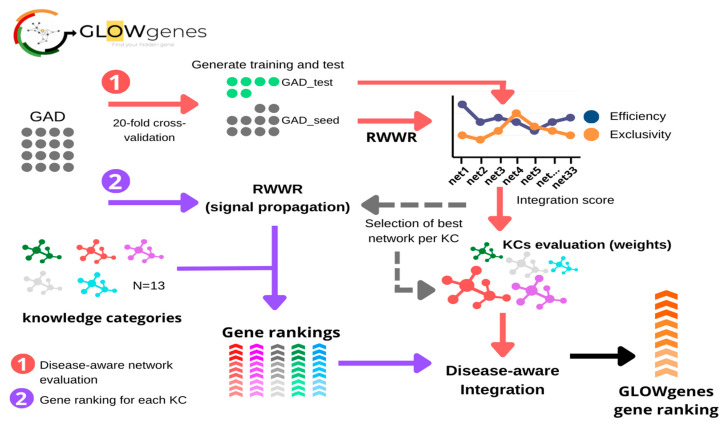
General schema of the GLOWgenes algorithm. GLOWgenes takes as input a set of genes associated with a disease (GAD) and performs two steps in parallel: (1) Step 1, the disease-aware network evaluation (red arrows), where GADs are randomly sorted into test (green dots) and training (grey dots). Efficiency (blue) and exclusivity (orange) are calculated for every knowledge category (KC) and an integration score is calculated. The integration scores merge efficiency and exclusivity obtained for the best performing KC network; (2) Step 2, gene ranking for each KC (purple arrows), where GADs are propagated in every best performing network (Step 1) using random walk with restart (RWWR) algorithm, producing a ranking of all genes for every KC (shown in different colors). KCs’ integration scores and KCs’ gene rankings are integrated into a GLOWgenes gene ranking as the KC performance-weighted average of normalized gene scores.

**Figure 5 ijms-24-01661-f005:**
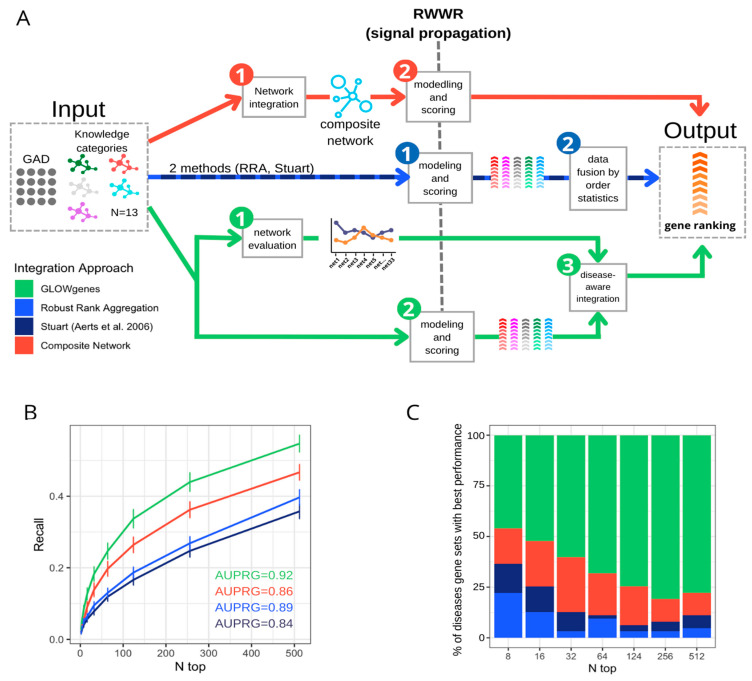
Benchmarking of three data integration approaches for disease candidate gene prediction. (**A**) General schema of three approaches for the integration of various datasets applied to the prioritization of phenotype-associated genes. In red, the composite network approach first merged the available networks into a composite network, (1) that is the subject of the modeling and scoring step (2) to generate the gene ranking. In blue, the data fusion approach applies first modeling and scoring in every dataset (1) to merge them into a unique ranking using order statistics (2). In green, GLOWgenes approach takes two steps, the evaluation of the performance of every network (1) and the modeling and scoring of the networks producing a ranking for each one (2). In a third step, (1) and (2) are merged to produce a single gene ranking (3). (**B**) Recall at various n-top of the performance of data integration methods. Precision–recall-gain curve associated area (AUPRG) is shown for every method. Methods are colored based on the types of approaches described in (**A**). (**C**) Percentage of diseases (PanelApp disease gene sets) for which each method performed best, measured at various n-top. Methods are colored based on the types of approaches described in (**A**).

**Figure 6 ijms-24-01661-f006:**
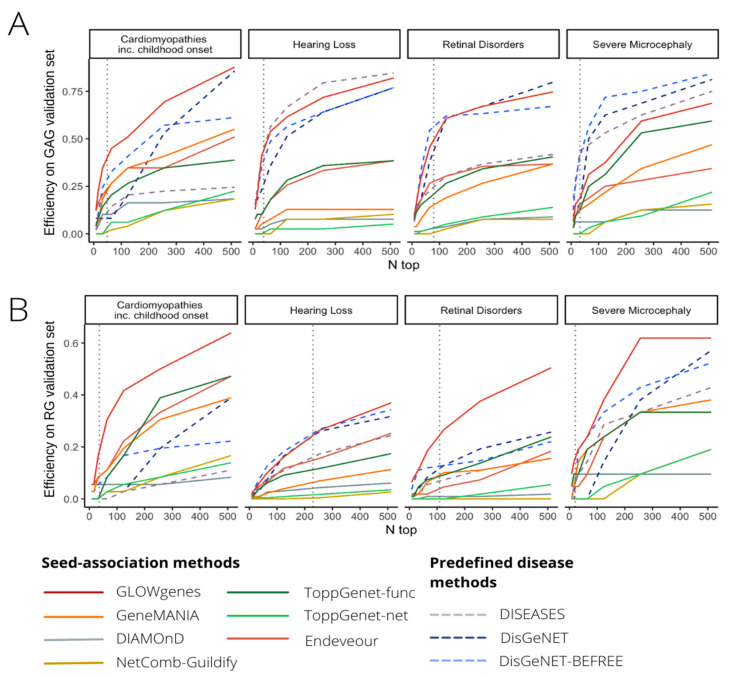
Benchmarking of tools for disease candidate genes prioritization. (**A**) Comparative evaluation of 10 tools for the prioritization of gene disease candidates using Green–Amber genes from PanelApp (GAGs, high evidence disease-associated genes) for validation in four selected diseases (pilot diseases). Recall levels at different n-top genes are represented. (**B**) Same to A, but using red genes from PanelApp (RGs, low-evidence disease-associated genes) as validation sets.

**Table 1 ijms-24-01661-t001:** Syndromic inherited retinal dystrophy candidate genes captured by GLOWgenes that hold pathogenic/likely pathogenic variants and fulfill phenotype in cases analyzed by WES. Variant pathogenicity is coded according to ACMG guidelines as: 1 (benign), 2 (likely benign), 3 (uncertain significance), 4 (likely pathogenic), and 5 (pathogenic).

Family	Gene	Variant	Consequence	CADD	Max POP Freq	Pathogenicity	Genotype	Inheritance
Case 1	SHH	NM_000193.2:c.676G>A	Missense variant	27.3	8.8 × 10^−05^	5	0/1	AD
Case 1	DNAH5	NM_001369.2:c.13486C>T	Stop gained	59	1.34 × 10^−04^	5	0/1	AR
Case 2	GLI1	NM_005269.2:c.762C>T	Splice region variant	15.05	6.67 × 10^−05^	4	0/1	NA

## Data Availability

GLOWgenes code is available at https://github.com/TBLabFJD/GLOWgenes. Networks used are available at https://figshare.com/s/75920a2cd985aa168fa2.

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
