# Peer review of "Prioritization of New Candidate Genes for Rare Genetic Diseases by a Disease-Aware Evaluation of Heterogeneous Molecular Networks"

_ijms, 2023, doi:10.3390/ijms24021661_

Round 1

Reviewer 1 Report

The authors have developed an algorithm that uses heterogeneous sources of biological information to prioritize disease-associated genes based on network approaches. The paper is clear and well-written and it is obvious that the authors have spent time generating figures to help the reader and potential users to understand their methodology. The methods are well described. The results look promising. 

I have no major issues with the manuscript, but my main concern is that I was not able to run the script provided at https://github.com/TBLabFJD/GLOWgenes

The code would be easier to use if the authors add a test folder, containing a gene panel example, and the network cfg file needed. And the command 

python GLOWgenes.py -i diseaseGenes.txt -n networks.cfg -o outputdir -p

could be replaced by one with the test data. 

When I try to run the script with one example panel, I get the following error

"GLOWgenes.py", line 112

    ",".join([str(graph.subgraph(c).number_of_nodes()) for c in nx.connected_components(graph)])]                                                                                             ^

SyntaxError: unmatched ']'

The script would also benefit if there was a more detailed error handling: checking the format of the genes file, or the existence of the input files (gene set and networks) and results folder

 There are absolute paths in the script 

with open("/home/proyectos/bioinfo/lorena/pSNOW/HGNC_GENES.txt", 'rt', encoding='utf-8') as hgnc:

I also would recommend mentioning the URL of the website in the abstract (www.glowgenes.org), to make the tool more visible. About the website, the link "Precomputed gene candidate prioritization for PanelApp disease genesets" is broken (it takes you to a 404 error in google drive).    

The button "Download Networks" takes you to the github repository https://github.com/TBLabFJD/GLOWgenes, where I do not see any network file. I believe it is important to make the networks available. 

Besides this, I only have a couple of questions: 

1. Could the better performance of the phenotype KC be explained by some kind of data leakage? To what extent are the gene-disease associations taken from the HPO annotations contained in the panel app gene sets? 

2. I find interesting the idea that different diseases would perform better in networks generated from different sources of information, and the selection step seems to be important to improve performance with respect to other methods. But I wonder, why not merge all datasets belonging to the same knowledge category, produce 13 networks, evaluate the performance in these 13 networks, and get the scores to do the final integration? What is the advantage of working with 33 networks? With the added disadvantage that some networks are really small, thus, will produce empty values, that will be necessary to impute. 

3. why the Restarting walk probability (rwp) was set up to 0.75? Isn't this a very high value? 

4. What method of imputation is for imputing scores for missing genes when producing the overall gene ranking of GLOWgenes. Why not just not taking into account those networks? 

I would also recommend adding the number of green/amber (GAG) and red genes (if any) to Table S2 with the List of 91 gene sets from the PanelApp resource used to diagnose genetic diseases. 

The link to the HPO file (https://hpo.jax.org/app/download/annotation) is broken. 

The way in which the HPOext network is constructed should be described in the methods section

Author Response

REVIEWER 1

The authors have developed an algorithm that uses heterogeneous sources of biological information to prioritize disease-associated genes based on network approaches. The paper is clear and well-written and it is obvious that the authors have spent time generating figures to help the reader and potential users to understand their methodology. The methods are well described. The results look promising.

I have no major issues with the manuscript, but my main concern is that I was not able to run the script provided at https://github.com/TBLabFJD/GLOWgenes

The code would be easier to use if the authors add a test folder, containing a gene panel example, and the network cfg file needed. And the command

python GLOWgenes.py -i diseaseGenes.txt -n networks.cfg -o outputdir -p

could be replaced by one with the test data.

We thank the reviewer for the positive feedback and agreed that a test example would be useful to show the usability of the program. We have added a test folder (https://github.com/TBLabFJD/GLOWgenes/tree/master/example) containing a README file with full instructions, including the command to be used and the explanation of the whole set of output files.

When I try to run the script with one example panel, I get the following error

"GLOWgenes.py", line 112

",".join([str(graph.subgraph(c).number_of_nodes()) for c in nx.connected_components(graph)])] ^

SyntaxError: unmatched ']'

We apologize for this bug that came from a previous version of GLOWgenes. Now it has been fixed.

The script would also benefit if there was a more detailed error handling: checking the format of the genes file, or the existence of the input files (gene set and networks) and results folder

We have added a more complete error handling to the code that we hope can make the usage more friendly.

There are absolute paths in the script

with open("/home/proyectos/bioinfo/lorena/pSNOW/HGNC_GENES.txt", 'rt', encoding='utf-8') as hgnc:

We again apologize for this oversight that it is now fixed.

I also would recommend mentioning the URL of the website in the abstract (www.glowgenes.org), to make the tool more visible. About the website, the link "Precomputed gene candidate prioritization for PanelApp disease genesets" is broken (it takes you to a 404 error in google drive).

We have added url to the abstract as requested, and also fixed the link of the precomputed results for 91 genetic diseases as defined by PanelApp: https://figshare.com/articles/dataset/Gene_Candidate_Prioritizations_In_Genetic_Diseases_With_GLOWgenes_zip/21770501.

The button "Download Networks" takes you to the github repository https://github.com/TBLabFJD/GLOWgenes, where I do not see any network file. I believe it is important to make the networks available.

The networks have now been released on figshare and the download button is now functional.

Besides this, I only have a couple of questions:

1. Could the better performance of the phenotype KC be explained by some kind of data leakage? To what extent are the gene-disease associations taken from the HPO annotations contained in the panel app gene sets?

This is a interesting point. As HPO definitions are quite close to genetic disease definitions, it was expected that networks based on phenotypes would perform well, at least in terms of recall. We have added this to the discussion. In this sense we believe that, in the case of rare genetic diseases, prioritization methods based only on HPO annotation can lead to obtain very reliable candidates but are probably not that suitable for the discovery of hidden associations. Interestingly, the phenotype network is not the best performing (even considering recall) in all the diseases, and the reasons might be various, including that a disease is not that well characterized in terms of HPO terms, or that the knowledge of genes causing those particular phenotypes is scarce. On the other hand, GLOWgenes could also be used for other kind of pathologies or groups of genes that are conceptually more distant to HPO definitions.

2. I find interesting the idea that different diseases would perform better in networks generated from different sources of information, and the selection step seems to be important to improve performance with respect to other methods. But I wonder, why not merge all datasets belonging to the same knowledge category, produce 13 networks, evaluate the performance in these 13 networks, and get the scores to do the final integration? What is the advantage of working with 33 networks? With the added disadvantage that some networks are really small, thus, will produce empty values, that will be necessary to impute.

This is also a good point. In contrast to others, our strategy was not to generate a composite (unique) network integrating the knowledge of several functional networks. This decision is based on the validation of our main hypothesis, that distinct networks would be of special interest in particular diseases in spite of their topological features, or the nature of the functional associations described. However, the reviewer is right in that the possibility to merge networks within knowledge categories (KCs) is there. We decided not to do so basically because although networks from some KCs like PPIs are quite homogeneous in their definitions, others, like genomic localization or coessentiality include different type of knowledge. On the top of this, networks within a KC might also have different technical features in their generation and different levels of precision in their conception, this is the case, for instance, of the KC regulation. These distinct features might be optimal to obtain good candidates for some diseases.

3. why the Restarting walk probability (rwp) was set up to 0.75? Isn't this a very high value?

Regarding rwp score, Vanunu et al. reported that the RWWR algorithm is not sensitive to the actual choice of this parameter as long as it is above 0.5. We did a small checking ourselves with same conclusion. It was setup to 0.75 as seen in other implementations (for instance: https://github.com/emreg00/guild/blob/master/src/random_walk.r). We have added this information to Materials and Methods “Network signal propagation” section.

Reference: Vanunu O, Magger O, Ruppin E, Shlomi T, Sharan R. Associating genes and protein complexes with disease via network propagation. PLoS Comput Biol. 2010 Jan 15;6(1):e1000641. doi: 10.1371/journal.pcbi.1000641. PMID: 20090828; PMCID: PMC2797085.

4. What method of imputation is for imputing scores for missing genes when producing the overall gene ranking of GLOWgenes. Why not just not taking into account those networks?

In fact we are very conservative in this sense. Gene score imputation is performed assigning to the genes not evaluated in a KC, the lowest normalized score obtained for any gene in that KC. Therefore, the effect should be the same as not including them. We have added this to Materials and Methods.

I would also recommend adding the number of green/amber (GAG) and red genes (if any) to Table S2 with the List of 91 gene sets from the PanelApp resource used to diagnose genetic diseases.

Number of genes have been added to the table.

The link to the HPO file (https://hpo.jax.org/app/download/annotation) is broken.

The reviewer is right, the link is no longer available and we have updated it.

The way in which the HPOext network is constructed should be described in the methods section

We apologize for this oversight, HPOext is the resultant network of imputing genes to HPO ancestors from their more specific annotation. HPO provides both levels of annotation. We have added this to Materials and Methods.

Reviewer 2 Report

The study of rare genetic diseases is an important field of medical research and genetics.

The authors "wanted to build a diverse and rich gene functional information framework to be 76 used for the prediction of new gene-disease associations." 

The article describes in detail the methods used. Although clustering is a well-established method, it is also useful to point out the more important elements of this statistical method.

Further questions and suggestion:

How they could interpret the differences observed in Fig2A and Fig2C.

A small suggestion is to write out very small p-values as unnecessary, in this case it is worthwhile to use the notation "<0.001" to describe significance.

Statistical analyses included the Wilcoxon singed rank test. One of their main criteria was the concordance of the variances. The Wilcoxon rank test tends to give significant results even if the shape of the two distributions is different. In this case, the test examines the dissimilarity of the distributions.

In the case of Fig2B, the line plot can be misleading, it might be worth using a different plot or simply using markers to represent efficiency. Perhaps use a different, more illustrative figure.

Having considered and responded to the above minor points and made the minor changes, I recommend the article for acceptance.

The article is well thought out, a result of complex research, which at the same time highlights also the importance of bitinformatics.

Author Response

REVIEWER 2

The study of rare genetic diseases is an important field of medical research and genetics.

The authors "wanted to build a diverse and rich gene functional information framework to be 76 used for the prediction of new gene-disease associations."

The article describes in detail the methods used. Although clustering is a well-established method, it is also useful to point out the more important elements of this statistical method.

Further questions and suggestion:

How they could interpret the differences observed in Fig2A and Fig2C.

While efficiency (Figure 2A) shows the capacity of a KC to recover genes associated to the disease, the exclusivity (Figure 2C) summarizes the ability of a KC to detect genes associated to the disease uniquely, so none (or few) other KCs can recover. As mentioned by the reviewer, these two graphs show different trends, regarding efficiency we observed that the KCs with networks based on phenotype co-annotations and on co-citations in scientific literature overtake the rest. In contrast, all KCs seem to have a role detecting genes uniquely in some disease. Thus, phenotype and co-citation seem to represent the kind of knowledge that are able to represent better the associations between genes involved in the same genetic mendelian disease. One of the possible explanations is that the genes not recovered by these two KCs are probably the latest discovered. This is supported by the comparison made in the performance of GLOWgenes selecting the training set of genes randomly and taking in consideration the publishing date of the gene-disease association. Thus, it is possible that apart from these two KCs, the rest have a more balanced contribution in recovering genes, as shown by the exclusivity (Figure 2C).

A small suggestion is to write out very small p-values as unnecessary, in this case it is worthwhile to use the notation "<0.001" to describe significance.

Thanks, we changed this in manuscript and figure S4.

Statistical analyses included the Wilcoxon singed rank test. One of their main criteria was the concordance of the variances. The Wilcoxon rank test tends to give significant results even if the shape of the two distributions is different. In this case, the test examines the dissimilarity of the distributions.

Wilcoxon is used to compared the distributions of the recall values in a particular family of diseases compared to the rest. Since the shape of the distributions was not checked, we used a non parametric test. Graphs in Figure S4 show a great difference between the two distributions compared for the three diseases reported to have a particular KC of special importance.

In the case of Fig2B, the line plot can be misleading, it might be worth using a different plot or simply using markers to represent efficiency. Perhaps use a different, more illustrative figure.

Thanks to the reviewer for pointing this out, we have now changed Figure 2B to be represented as a heatmap, as a more natural way of representing a ranking.

Having considered and responded to the above minor points and made the minor changes, I recommend the article for acceptance.

The article is well thought out, a result of complex research, which at the same time highlights also the importance of bitinformatics.